# Effect of virtual group counseling based on health literacy on the empowerment and self-care of pregnant women: A randomized controlled trial

Zahra Rezaee[1], Fatemeh Bakouei [2*], Hajar Adib-Rad[2], Soraya Khafri[3], Zeinab Gholamnia Shirvani[3], Farzaneh Khorsand[4]

1 Student Research Committee, Babol University of Medical Sciences, Babol, I.R. Iran, 2 Infertility and Reproductive Health Research Center, Health Research Institute, Babol University of Medical Sciences, Babol, I.R. Iran, 3 Social Determinants of Health Research Center, Health Research Institute, Babol University of Medical Sciences, Babol, I.R. Iran, 4 Guilan University of Medical Sciences, Guilan, Iran

* bakouei2004@yahoo.com

## Abstract

### Introduction

Pregnancy presents a valuable opportunity for education and counseling aimed at enhancing the health outcomes of both mothers and newborns. This study was conducted to assess the effectiveness of virtual group counseling based on health literacy in promoting empowerment and self-care among pregnant women.

### Materials and methods

This open label, randomized controlled trial was conducted in 2024-2025 involved 84 pregnant women. Participants who met the inclusion criteria were randomly allocated to either the intervention or control group through a block randomization method. In addition to receiving routine prenatal care, the intervention group participated in weekly virtual group counseling sessions focused on health literacy. Standardized questionnaires were used to assess empowerment and self-care as primary outcomes, and health literacy as a secondary outcome. These assessments were conducted both before the intervention and four weeks after its completion. The statistical analysis followed the intention-to-treat principle. To compare the changes in outcomes over time between the two groups, the generalized estimating equations model was employed.

### Results

The mean differences in empowerment, self-care, and health literacy scores before and after the intervention in the intervention group were as follows: 16.68 (95% confidence interval: 14.39, 18.95), 29.78 (95% confidence interval: 24.24, 35.19), and 19.37 (95% confidence interval: 16.71, 22.32), respectively (P<0.001). While the

**Data availability statement:** All relevant data are within the manuscript and its Supporting Information files.

**Funding:** The author(s) received no specific funding for this work.

**Competing interests:** The authors have declared that no competing interests exist.

changes in the scores of these three outcomes in the control group were 1.93 (95% confidence interval: 1.12, 2.80), 2.52 (95% confidence interval: 1.23, 4), and 1.85 (95% confidence interval: 1.60, 3.25), respectively. These between-group differences were statistically significant (P<0.001), highlighting the intervention's effectiveness.

## Conclusion

The virtual group counseling intervention based on health literacy significantly improved empowerment, self-care, and health literacy scores among pregnant women. This approach is recommended to health program planners as an effective strategy for empowering pregnant women and promoting better maternal health outcomes.

## Trial registration

This trial was registered at the Iranian Registry of Clinical Trials with the identifier IRCT20221109056451N1.

## Introduction

Pregnancy is a critical period of biological, physical, and hormonal changes [1]. Proper care during pregnancy can reduce the risk of mortality and complications associated with pregnancy and childbirth [2].

Self-care during pregnancy refers to the choices and activities adopted by a pregnant woman to manage health-related problems and concerns throughout this critical period, with the objective of optimizing maternal and fetal well-being [3]. The dimensions of prenatal self-care include adequate nutrition, regular physical activity, personal hygiene, adherence to routine prenatal care, and avoidance of tobacco, alcohol, and other harmful substances [4]. Engaging in positive self-care practices has been consistently associated with improved maternal and neonatal outcomes and a reduced incidence of adverse pregnancy complications [5].

Empowering pregnant women to enact effective self-care behaviors is essential for safeguarding their health and that of their newborns. Such empowerment hinges on the acquisition of relevant knowledge, skills, and resources [6]. According to the World Health Organization (WHO), empowerment is a dynamic process through which individuals gain greater control over decisions and actions influencing their health and well-being [3].

Strengthening both empowerment and self-care during pregnancy can significantly mitigate the risk of complications and support a healthier gestational experience [7]. Central to this process is an adequate level of health literacy—the capacity to access, understand, appraise, and apply health-related information to make informed decisions. Health literacy serves as a foundational component of empowerment, enabling pregnant individuals to actively participate in health-promoting behaviors for themselves and their children. In the absence of sufficient health literacy, the ability to engage in informed decision-making regarding care options and health outcomes

becomes markedly constrained [8,9]. Empirical evidence further indicates a positive association between health literacy and various dimensions of empowerment during pregnancy [6]. Consequently, targeted educational interventions and counseling strategies should prioritize the enhancement of health literacy as a key mechanism for fostering self-care and empowerment [10].

Systematic reviews reveal that the majority of existing research on health literacy and related interventions has focused on women of reproductive age who are not pregnant [11,12]. There remains a notable gap in evidence concerning pregnant populations, particularly in diverse geographic and socioeconomic contexts. Furthermore, studies conducted across various regions of Iran indicate that levels of empowerment and health literacy among pregnant women remain modest and limited [7,13,14]. Addressing this gap, the present study aims to evaluate the effectiveness of a group-based virtual counseling intervention, grounded in health literacy principles, on the empowerment and self-care behaviors of pregnant women.

This research distinguishes itself from previous studies by employing a virtual counseling intervention that focuses on general aspects of health literacy and employs a multidimensional strategy to influence empowerment and self-care outcomes during pregnancy. If proven effective, this intervention could be integrated into standard prenatal care, enhancing women's self-care and empowerment during pregnancy and childbirth, and ultimately improving maternal and infant health outcomes.

## Materials and methods

### Study design

This randomized controlled trial (RCT) was prospectively registered in the Iranian Registry of Clinical Trials with the identifier number IRCT20221109056451N1. This RCT with two parallel arms (control and intervention) conducted with block randomization method using blocks of four was employed following a 1:1 allocation ratio. The trial follows the guidelines of Consolidated Standards of Reporting Trials (CONSORT). The study protocol was published in journal of BMJ Open with https://bmjopen.bmj.com/content/bmjopen/15/7/e097991.full.pdf in July 2025 [15].

### Participants

Participants for this study were recruited from the comprehensive urban-rural health service centers in Langrod city (in Northern Iran). The inclusion and exclusion criteria were shown in Table 1. Eligible pregnant women were included in the study using a convenience sampling method after reviewing the inclusion and exclusion criteria and obtaining written informed consent.

Table 1. Inclusion and exclusion criteria.

| Inclusion criteria | Exclusion criteria |
| --- | --- |
| • Gestational age between 14–20 weeks | • High-risk pregnancy conditions |
| • A health literacy score below 66 (indicating limited health literacy) | • Known chronic diseases |
| • At least middle school education level | • Self-reported mental illness |
| • Ownership and ability to operate a smartphone | • Educational backgrounds in medical or paramedical fields |
| • Proficiency in Persian language | • Substance dependence |
| • Voluntarily participate in the trial and sign the informed consent. | • Prior formal empowerment and self-care training |
| Candidates who meet all the above criteria will be included | Candidates meeting any of the above criteria will be excluded |

## Sample size

The sample size was calculated based on the primary outcomes using G Power software version 3.1.9.2. Following the framework established by Houshmandpour et al. [7], the effect sizes for the variables of empowerment and self-care were determined to be 0.68 and 0.71, respectively, at a confidence level of 0.95 and a power of 80%. This analysis revealed that a minimum sample size of 70 and 66 participants would be necessary to detect significant differences between the two groups at the specified time points. Taking into account an expected dropout rate of 20%, the actual required sample size of 84 was established, with 42 pregnant women allocated to each group.

## Randomization and blinding

Eligible participants were randomly allocated to the groups utilizing a permuted-block randomization method with four blocks, maintaining an equal one-to-one ratio. This technique prevents the researcher from anticipating the subsequent group assignment for each participant. The randomization list was created using the website (http://random.org) and was developed by the team's statistics expert colleague. To reduce potential selection bias in the study, this list was accessible only to the study supervisor.

Full blinding was not feasible due to the nature of the virtual group counseling intervention—an inherent limitation in many non-pharmacological trials, particularly those involving participatory or behavioral interventions [16]. However, consistent with current methodological recommendations, we implemented several strategies to mitigate potential bias: (1) outcome assessors and data analysts were blinded to group allocation through the use of coded datasets; (2) all outcomes were measured using validated, self-administered questionnaires to minimize interviewer bias; and (3) group allocation and participant contact information were stored separately to preserve blinding during follow-up. These measures align with best practices for minimizing detection and performance bias in open-label trials where participant or provider blinding is not possible.

## Interventions

The intervention contents based on a health literacy approach related to empowerment and self-care, were developed in accordance with national guidelines, as well as literature reviews. The content validity of the drafts was assessed by the research team and five reproductive health professionals from outside the team. Weekly counseling sessions began at 20 weeks of pregnancy, consisting of five 30–40-minute sessions held in groups of five to eight participants at a set time. An audio-video platform that could be readily installed on smartphones was used to conduct group counseling online.

Group counseling was conducted using the Zoom application. The session link was sent to the participants in advance, and they were able to join by just clicking it and connecting to the internet. Team supervisors oversaw the group members' interactions during the sessions, as well as their interactions with the first researcher.

The counseling intervention components were designed as a staged program intended to address multiple dimensions of health literacy (access to information, reading, understanding, appraisal, and decision-making or behavioral intention) to foster empowerment and self-care. The initial and second sessions focused on empowerment dimensions; enhancing self-efficacy, future image, self-esteem, perception of support from others, and pleasure of adding a new member to the family. The subsequent three sessions concentrated on self-care, targeting nutrition practices and physical activity, routine prenatal care, personal hygiene, and smoking and drug use. In the final session also summarized the overall content and reinforced key messages. Collectively, this multi-domain approach seeks to develop health-decision competencies and to influence empowerment and behavioral outcomes relevant to maternal health. This explanation is provided in Table 1 in the published protocol of this article [15]. The intervention group received more information as necessary, while erroneous information was corrected and accurate knowledge was reinforced.

The pregnant women in the control group received only routine prenatal care according to national guidelines. To ensure the control group gained from the intervention, the counseling material was distributed through social messenger after the study.

## Outcome measures

The primary outcomes of this study included changes in the mean scores of empowerment and self-care, while the secondary outcome involved changes in the mean health literacy scores of pregnant women. The questionnaires assessing health literacy, empowerment of pregnant women, and self-care during pregnancy were collected at baseline and four weeks after the final counseling session for the intervention group and at the equivalent time point for the control group, specifically during weeks 28–30 of gestation.

**Empowerment.** Empowerment was evaluated using the pregnant women empowerment questionnaire, including 27 items distributed over five domains: self-efficacy (6 items), self-esteem (7 items), future image (6 items), pleasure of adding a new member to the family (4 items), perception of support from others (4 items). Each item is rated a score from 1 to 4 on a 4-point scale, representing ranging from total disagreement to total agreement. A score ranging from one (the lowest) to four (the highest). The total score ranges from 27–108. In Iran, the questionnaire's face validity and content validity were assessed, and its Cronbach's alpha was found to be 0.84 [17].

**Self-care.** Self-care status was measured by the Pregnancy Self-Care Questionnaire, which is a 57-item scale across 6 domains: nutrition performance (15 items), personal hygiene (23 items), smoking and drug use (4 items), exercise and physical activity (4 items), and routine pregnancy care (11 questions). A Likert scale with four response options, ranging from never (0) to always (3), is used to capture answers. The higher score, the better a woman takes care of herself. The questionnaire's reliability was assessed in Iran, and its Cronbach's alpha was determined to be 0.86 [4].

**Health literacy.** Health literacy was measured by the Health Literacy for Iranian Adults questionnaire (HELIA-33) with five dimensions: access, reading, understanding, appraisal, and decision-making/behavioral intention. The range of possible scores is 0–100. The total health literacy score is divided into two categories: "limited health literacy: score 66 or below" and "desirable health literacy: score above 66". In Iran, the validity of this questionnaire has been established with a Cronbach's alpha of 0.89 [18].

**Demographic and fertility questionnaire.** Based on the literature review, we pre-specified a set of baseline sociodemographic and reproductive characteristics potentially associated with the study outcomes—empowerment, self-care, and health literacy. These variables included age, education, employment status, income, parity, history of abortion, pregnancy planning status, and primary source of health information.

## Data collection

The implementation of this study commenced in June 2024 and continued until November 30. The study was prospectively approved by the Ethics Committee of Babol University of Medical Sciences (IR.MUBABOL.HRI.REC.1402.202) and registered on the Iranian Registry of Clinical Trials with confirmation code IRCT20221109056451N1. The first researcher met with pregnant women who referred to six comprehensive health service centers to receive routine prenatal care. These centers affiliated with Gillan University of Medical Sciences in Langrod city, located in Northern Iran.

Following a detailed introduction of the research objectives and procedures, the researcher obtained the written informed consent before proceeding with data collection. Comprehensive medical histories were reviewed to identify eligible pregnant women meeting the study's inclusion criteria, also a health literacy score of ≤66 (limited health literacy) according to the Iranian Adult Health Literacy Questionnaire. Consequently, eligible participants completed the questionnaires to assess variables of sociodemographic and fertility, empowerment, and self-care. At this stage of the study, participants were randomly allocated to either the intervention or control group. Follow-ups and data collection were completed by the end of November 2024 (Fig 1 illustrates the study flow diagram).

## Statistical analysis

Data analysis was done using the Statistical Package for Social Sciences (version 22). Categorical variables (descriptive characteristics) were reported as numbers and percentages. Continuous variables are presented as mean and standard

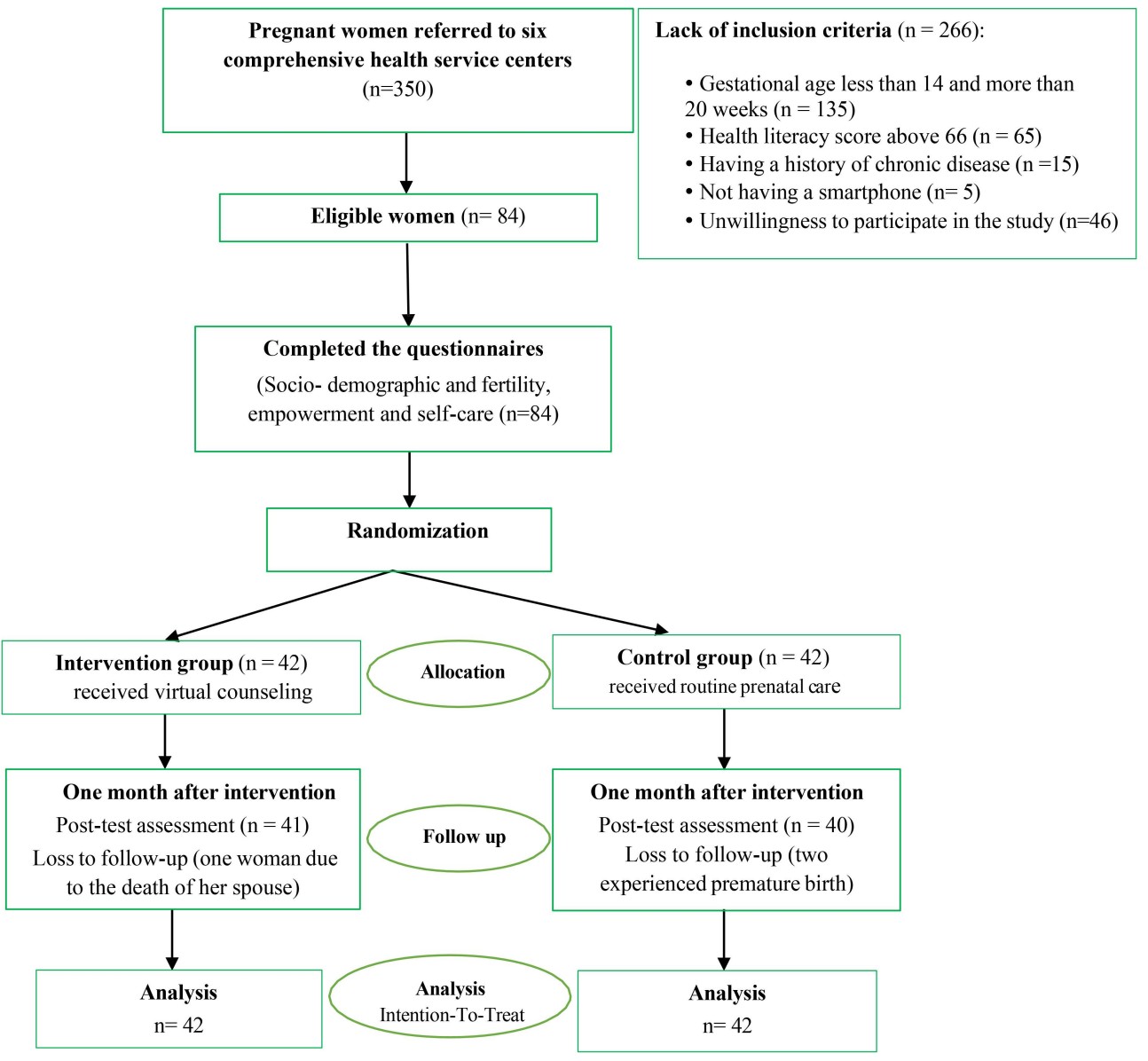

**Fig 1. Study flow diagram.**

deviation (SD). For normally distributed variables, we employed Student's t-test, while categorical data were compared using either the chi-square test. All analyses followed the intention-to-treat principle, including all randomized participants in their originally assigned groups irrespective of intervention adherence. To compare longitudinal changes in outcome measures between study groups, we applied generalized estimating equations (GEE) models. Results are reported with 95% confidence intervals (CIs), and statistical significance was set at $p < 0.05$ for all tests.

## Ethical considerations

This study was registered in the Iranian Registry of Clinical Trials and received ethical approval from the Research Ethics Committee of Babol University of Medical Sciences (Approval Code: IR.MUBABOL.HRI.REC.1402.202). All participants

voluntarily agreed to participate in the study by providing a written informed consent prior to enrollment. The research adhered to the principles outlined in the Declaration of Helsinki. Following completion of the intervention study, participants in the control group received equivalent counseling material to those distributed in the intervention group.

## Results

### Participant's characteristics

Eligibility screening was performed for 350 pregnant women, resulting in 220 ineligibles and 46 who declined participation. A total of 84 participants were randomly allocated to either the virtual counseling group (n = 42) or the control group (n = 42), as illustrated in Fig 1 (study flow diagram). The dropout of the study was 3.6%; one woman in the intervention group (resulting from the death of her spouse) and two women in the control group (experienced premature birth). The socio-demographic and fertility characteristics of the women included in this study are presented in Table 2. The study

Table 2. Participant's sociodemographic-fertility characteristics.

| Characteristics | Frequency (%) | | P-value chi-square tests |
|---|---|---|---|
| | Intervention (n = 42) | Control (n = 42) | |
| **Age (years)** | | | |
| ≥ 19 | 1 (2.4) | 2 (4.8) | 0.105 |
| 20-35 | 25 (59.5) | 29 (69.0) | |
| ≥ 35 | 16 (38.1) | 11 (26.2) | |
| **Level of education** | | | |
| Elementary/middle school | 5 (11.9) | 7 (16.7) | 0.788 |
| High school/diploma | 22 (52.4) | 22 (52.4) | |
| Academic | 15 (35.7) | 13 (31.0) | |
| **Job** | | | |
| Housewife | 38 (90.5) | 32 (76.2) | 0.079 |
| Employed | 4 (9.5) | 10 (23.8) | |
| **Family income** | | | |
| Sufficient | 5 (11.9) | 6 (14.3) | 0.400 |
| Relatively sufficient | 30 (71.4) | 33 (78.6) | |
| Insufficient | 7 (16.7) | 3 (7.1) | |
| **Parity** | | | |
| 0 | 16 (38.1) | 25 (59.5) | 0.131 |
| 1 | 20 (47.6) | 12 (28.6) | |
| 2 | 6 (14.3) | 5 (11.9) | |
| **History of abortion** | | | |
| Yes | 9 (21.4) | 3 (7.1) | 0.061 |
| No | 33 (78.6) | 39 (92.9) | |
| **Planned pregnancy** | | | |
| Yes | 32 (76.2) | 33 (78.6) | 0.794 |
| No | 10 (23.8) | 9 (21.4) | |
| **The most common source of health information** | | | |
| Ask from doctor and healthcare providers | 15 (35.7) | 18 (42.9) | 0.772 |
| Book, Internet, and social media | 14 (31.0) | 9 (21.4) | |
| Experiences of family and friends | 1 (2.40) | 1 (2.40) | |
| More than one option | 12 (30.9) | 14 (33.3) | |

population had a mean age of 30.33 ± 7.68 years. Baseline comparisons revealed no statistically significant differences between groups regarding socio-demographic and fertility characteristics, indicating successful randomization (P > 0.05).

## Changing of primary and secondary outcomes after intervention

The findings presented in Table 3 indicate that the intervention significantly improved the total scores of the empowerment, the self-care, and the health literacy among pregnant women in the intervention group (p < 0.001). Prior to the intervention, no statistically significant differences were observed in the overall scores of the three outcome variables between the control and intervention groups. However, following the intervention, the mean score changes in empowerment, self-care, and health literacy in the intervention group were 16.68, 29.78, and 19.37, respectively, which were significantly higher than those in the control group (1.93, 2.52, and 1.85, respectively) (p < 0.001). Notably, in addition to the overall scores, all dimensions of these three outcomes also improved significantly (accessible in S1-S3 Figs).

## Discussion

The present study was a randomized clinical trial conducted with the aim of investigating the impact of virtual group counseling based on health literacy on the empowerment and self-care of pregnant women. The results regarding the primary outcomes of the study indicated that the virtual group counseling intervention based on health literacy had a significant positive effect on increasing mean scores of the empowerment and self-care among participating pregnant women.

In line with the findings related to empowerment, the empowerment score after the intervention was significantly higher in the intervention group compared to the control group. In other words, the findings demonstrated that the virtual group counseling based on health literacy effectively enhanced all dimensions of empowerment of pregnant women in the intervention group. The greatest observed change was in the "future image" dimension (mean = 3.66), which is notable. These results highlight the importance and effectiveness of health literacy-based approaches in promoting the empowerment of pregnant women.

To the best of our knowledge and based on a comprehensive review of existing literature, no previous study has specifically investigated the impact of a health literacy–based approach on the empowerment of pregnant women. While several

Table 3. Comparison of the outcomes' scores before and after intervention in the groups.

| Outcomes | Time | Intervention group (Mean ±SD*) | Control group (Mean ±SD*) | P-value |
|---|---|---|---|---|
| Empowerment* | Before | 69.76 ± 10.33 | 71.52 ± 7.59 | 0.376 |
| | After | 86.56 ± 8.77 | 73.58 ± 6.25 | <0.001 |
| | Mean Difference (95% Confidence Interval) | 16.68 (14.39,18.95) | 1.93 (1.12,2.80) | <0.001 |
| | P-value$^\epsilon$ | | | <0.001 |
| Self-care** | Before | 96.00 ± 28.35 | 102.02 ± 16.78 | 0.239 |
| | After | 126.15 ± 20.76 | 105.13 ± 14.34 | <0.001 |
| | Mean Difference (95% Confidence Interval) | 29.78 (24.24,35.19) | 2.52 (1.23,4.00) | <0.001 |
| | P-value$^\epsilon$ | | | <0.001 |
| Health literacy*** | Before | 56.30 ± 11.25 | 57.46 ± 9.51 | 0.612 |
| | After | 75.91 ± 8.44 | 59.87 ± 8.90 | <0.001 |
| | Mean Difference (95% Confidence Interval) | 19.37 (16.71,22.32) | 1.85 (1.60,3.25) | <0.001 |
| | P-value$^\epsilon$ | | | <0.001 |

*The minimum and maximum possible scores were 27–108.

**The minimum and maximum possible scores were 0–171.

***The minimum and maximum possible scores were 0–100.

$^\epsilon$The P-Value indicates the interaction effect of time and group in the GEE model.

studies have explored group-based education, counseling, and care models using different methodologies, their findings are largely consistent with those of the present study.

Malchi et al. (2023) conducted a study examining the effect of group prenatal care on the empowerment of adolescent pregnant women and reported a statistically significant increase in empowerment scores in the intervention group compared to the control group [19]. In the current study, group counseling delivered through a health literacy–oriented approach led to an increase of 16.68 in the mean empowerment score. In contrast, Malchi et al. reported a mean increase of 8.17 in the empowerment score. Notably, both studies utilized the same 27-item Pregnant Women's Empowerment Questionnaire to assess empowerment levels.

This consistency in measurement tools, particularly within educational and counseling contexts, may enhance the validity and generalizability of findings across these studies. The greater improvement observed in the current study could be attributed to the integration of health literacy principles into the counseling framework, which may have facilitated a deeper understanding and application of health-related knowledge among participants.

In another study, Houshmandpour et al. investigated the impact of midwife-led group counseling using the Orem self-care model on self-care behaviors and empowerment among primiparous pregnant women. The results demonstrated statistically significant differences in empowerment scores between the two groups at three time points: pre-intervention, immediately post-intervention, and three weeks after the intervention [7]. In Houshmandpour's study, the group counseling sessions were conducted in person, whereas the present study aimed to facilitate access for mothers and reduce the need for face-to-face visits by adopting a virtual mode of education. This innovative and flexible approach holds particular significance in contemporary times, especially in light of potential challenges such health and social crises. It enables mothers to engage in learning and enhance their competencies at convenient times that align with their personal circumstances.

These findings underscore the importance and necessity of implementing educational and counseling interventions to empower pregnant women. Moreover, they highlight the growing need for novel and transformative approaches in maternal healthcare. In this context, integrating virtual, health literacy–based group counseling into routine prenatal care offers a scalable, equitable, and evidence-informed strategy to strengthen maternal outcomes, particularly in settings with limited access to in-person services or specialized providers.

Another primary outcome of the present study was the assessment of self-care behaviors using the Pregnancy Self-Care Questionnaire. The findings revealed that, following the counseling intervention, the mean score all self-care domains in the intervention group was significantly higher than that in the control group. The largest change occurred in the "personal hygiene" dimension, with a mean of 56.11. This result clearly indicates a substantial impact of virtual group counseling based on health literacy in improving self-care among pregnant women. These findings are consistent with those of Solhi et al. (2018), who also reported significant improvements in self-care following structured educational interventions during pregnancy [2].

Despite the similarity in results, the present study differs from Solhi's research in several key aspects. The most important difference lies in the design and approach to the intervention. To the best of our knowledge, this is the first study to implement a comprehensive general health literacy–based group counseling approach. In other words, all dimensions of health literacy were systematically addressed throughout the intervention. In contrast, Solhi et al.'s study focused only on domains specifically related to pregnancy-related health literacy. Such a holistic approach may offer benefits beyond the prenatal period, potentially influencing postpartum care and other stages of reproductive life. However, further research is needed to confirm these potential long-term effects. Specifically, longitudinal studies with follow-up periods extending through the postpartum phase (e.g., 6 weeks, 6 months, and 1 year after delivery) are needed to assess the durability of gains in empowerment, self-care, and health literacy.

In addition, Solhi's study utilized a researcher-developed self-care questionnaire consisting of 21 items, categorized into two domains: physical health and psychological well-being. In contrast, the present study employed a more comprehensive and standardized measurement tool, which may contribute to greater accuracy and reliability in assessing

self-care behaviors among pregnant women. Furthermore, Solhi's intervention was delivered through face-to-face educational sessions, whereas the current study adopted a virtual group counseling approach. This shift in delivery method not only facilitates easier access to scientific resources and emotional support for expectant mothers but also provides an effective solution for enhancing optimal engagement and social support in today's increasingly digital world.

In this study, changes in the mean health literacy scores of pregnant women following a virtual group counseling intervention based on health literacy were also examined (secondary outcome). Our results demonstrated that, after the counseling intervention, the mean health literacy score in the intervention group was significantly higher than that in the control group. The greatest change was observed in the dimension of "appraisal" (with a mean of 22.78). This finding aligns with the study by Aslantekin Ozcoban et al., which investigated the effect of an intervention on health literacy among 220 pregnant women using the Turkish Health Literacy Scale (THLS-32). Their results clearly confirmed the effectiveness of health literacy–oriented education in improving health literacy levels among pregnant women [20]. Both studies provide evidence that preventive educational and counseling interventions can effectively enhance pregnant women's ability to understand and manage their health.

In another study by Kharazi et al., which aimed to evaluate the impact of an educational intervention based on Bandura's theory of self-efficacy on improving health literacy levels and pregnancy outcomes, findings showed that the theory-based intervention led to a significant increase in mean health literacy scores as well as improvements in neonatal birth weight. These results reinforce the importance of theory-driven educational approaches—particularly those emphasizing self-efficacy—in promoting health literacy and maternal and child health outcomes [21].

A key distinction between the present study and Kharazi's lies in the method of intervention delivery. In our study, virtual group counseling was employed, offering the benefits of interactive engagement and shared learning opportunities. In contrast, Kharazi's study utilized face-to-face group discussions and question-and-answer sessions. It appears that both approaches can be effective in enhancing the health literacy of pregnant women, albeit through different modalities.

Overall, the findings of the present study indicate that a virtual group counseling approach based on health literacy can effectively exert a positive influence on all dimensions of health literacy—including reading, accessing, understanding, evaluating, and applying health-related information. Moreover, existing evidence suggests that higher levels of health literacy may lead to better health outcomes by increasing awareness and knowledge, strengthening communication skills with healthcare providers, and promoting active participation in health-related decision-making (23).

In general, pregnancy is widely recognized as a critical window of opportunity for health education and behavioral change. According to findings from various studies, most women demonstrate high motivation to engage in self-care and adopt healthier behaviors during pregnancy in order to achieve favorable outcomes for both themselves and their infants. At the same time, due to the large volume of health-related information they receive during this period, pregnant women require specific skills to access, understand, and evaluate credible health information, and to make informed decisions about its application at the appropriate time. These competencies are essential for recognizing warning signs and preventing adverse pregnancy outcomes. In this sensitive phase, during which women experience significant physical, psychological, and social changes, having an adequate level of health literacy appears to be crucial. Inadequate understanding of health care recommendations hinders informed decision-making and may lead to poor health outcomes. Health literacy is therefore considered a key component in empowering pregnant women to effectively interact with the healthcare system and engage in self-care practices that promote the health of both mother and child.

## Limitation

Due to the nature of the intervention and the study design, blinding was not feasible, which may have influenced the final results. To minimize potential bias and errors, several methodological strategies were performed during the design and implementation phases, as detailed in the methodology section.

## Conclusion

The findings of the present study demonstrate the positive impact of a virtual group counseling intervention based on health literacy on the outcomes of interest. The results indicate that this intervention significantly improved the mean scores of the empowerment, self-care, and health literacy among pregnant women. These findings clearly suggest that implementing a health literacy–based group counseling approach—particularly through virtual platforms—can enhance the capacity of pregnant women to engage in effective self-care practices by improving their level of health literacy. Given the numerous challenges faced by women during pregnancy, such interventions can serve as a valuable tool for promoting maternal and neonatal health. Therefore, special emphasis should be placed on the development and expansion of similar counseling programs aimed at empowering pregnant women, particularly during high-risk or vulnerable periods. Such initiatives have the potential to make a significant contribution to improving pregnancy outcomes and overall public health.

## Supporting information

**S1 Fig. Dimensions of empowerment.**
(TIF)

**S2 Fig. Dimensions of prenatal self-care.**
(TIF)

**S3 Fig. Dimensions of health literacy.**
(TIF)

## Acknowledgments

This work has extracted from the thesis related to Master's Degree of midwifery counseling was supported by Babol University of Medical Sciences. The authors hereby thank the honorable Research Vice-Chancellor of Babol University of Medical Sciences. We would like to express our thanks to the study participants for their enthusiastic cooperation.

## Author contributions

**Conceptualization:** Zahra Rezaee, Fatemeh Bakouei, Hajar Adib-Rad, Farzaneh Khorsand.

**Data curation:** Zahra Rezaee, Fatemeh Bakouei.

**Formal analysis:** Zahra Rezaee, Soraya Khafri.

**Investigation:** Zahra Rezaee, Fatemeh Bakouei, Zeinab Gholamnia Shirvani.

**Methodology:** Zahra Rezaee, Fatemeh Bakouei, Soraya Khafri.

**Project administration:** Zahra Rezaee, Fatemeh Bakouei, Zeinab Gholamnia Shirvani.

**Software:** Zahra Rezaee, Soraya Khafri.

**Supervision:** Fatemeh Bakouei.

**Writing – original draft:** Zahra Rezaee, Fatemeh Bakouei, Hajar Adib-Rad, Soraya Khafri, Zeinab Gholamnia Shirvani, Farzaneh Khorsand.

**Writing – review & editing:** Zahra Rezaee, Fatemeh Bakouei, Hajar Adib-Rad, Soraya Khafri, Zeinab Gholamnia Shirvani, Farzaneh Khorsand.

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
