## [Decision Letter · Decision Letter 0]

3 Nov 2025

Dear Dr. Bakouei,

Thank you for submitting your manuscript to PLOS ONE. After careful consideration, we feel that it has merit but does not fully meet PLOS ONE’s publication criteria as it currently stands. Therefore, we invite you to submit a revised version of the manuscript that addresses the points raised during the review process.

Thank you for submitting your manuscript. Both reviewers see merit in the work, but major revisions are required for acceptance, as both raise significant, non-negotiable points. There are no conflicts between the reviews; the concerns are complementary, and all required changes noted by both reviewers must be addressed. My primary concern, aligning with Reviewer 2, is the scholarly integrity and logical structure of the Introduction. As R2 meticulously documented, the Introduction currently suffers from severe and unacceptable citation issues (e.g., the inappropriate Ebola [Ref 1] and UHC [Ref 9] references); you are required to conduct a thorough audit and replace all mismatched citations with authoritative, primary sources. Concurrently, the Introduction must be substantially rewritten (R2) to establish a clear logical flow (global regional local gap) and to better conceptualize the links between empowerment, self-care, and health literacy. Furthermore, you are required to address all points from Reviewer 1: specifically, you must verify the manuscript against your published BMJ Open protocol for consistency, must provide the scoring details for Table 4, and must elaborate on the statistical relationships from Table 3 in the Methods and Discussion. Finally, the Discussion must be deepened (R2) to include a theoretical discussion of mechanisms and provide stronger, specific policy implications. A *recommendation* is to add the methodological citation for blinding suggested by R2, but the structural and citation revisions are the priority required for reconsideration.

Please submit your revised manuscriptDec 18 2025 11:59PM. If you will need more time than this to complete your revisions, please reply to this message or contact the journal office at plosone@plos.org . A rebuttal letter that responds to each point raised by the academic editor and reviewer(s). You should upload this letter as a separate file labeled 'Response to Reviewers'.A marked-up copy of your manuscript that highlights changes made to the original version. You should upload this as a separate file labeled 'Revised Manuscript with Track Changes'.An unmarked version of your revised paper without tracked changes. You should upload this as a separate file labeled 'Manuscript'.

We look forward to receiving your revised manuscript.

Kind regards,

Fatemeh Zarei, PhD

Academic Editor

PLOS ONE

2. Please include a separate caption for each figure in your manuscript.

Additional Editor Comments (if provided):

Reviewers' comments:

Reviewer's Responses to Questions

**Comments to the Author**

1. Is the manuscript technically sound, and do the data support the conclusions?

Reviewer #1: Yes

Reviewer #2: Yes

2. Has the statistical analysis been performed appropriately and rigorously?

Reviewer #1: Yes

Reviewer #2: I Don't Know

3. Have the authors made all data underlying the findings in their manuscript fully available?

Reviewer #1: Yes

Reviewer #2: Yes

4. Is the manuscript presented in an intelligible fashion and written in standard English?

Reviewer #1: Yes

Reviewer #2: Yes

Reviewer #1: Dear corresponding author

Your manuscript has been reviewed, some points can be revised.

Given that this research work has previously been published under a protocol in the journal BMJ Open with doi:10.1136/bmjopen-2024-097991, different parts of the manuscript should be reviewed and revised for any dissimilarity.

In Table 4, explain the details of the scores for the different dimensions of empowerment, self-care, and health literacy.

Provide a brief explanation of the statistically significant relationship between the characteristics listed in Table 3 with Empowerment, Self-care, and Health literacy outcomes in the Methods part and a comparison with other studies in the Discussion part.

Reviewer #2: Introduction

It lacks a clear logical progression from global → regional → local gap → aim. Several concepts (empowerment, self-care, health literacy) overlap heavily; the reader gets definitions but not a clear hierarchy among them. The research gap and rationale for the specific intervention (virtual group counseling) appear only in the final sentence — too sudden and not sufficiently prepared. Paragraphs are long, descriptive, and somewhat repetitive (definitions and effects are stated multiple times). The transition between empowerment → self-care → health literacy is mechanical rather than conceptual.

Please assess whether the Introduction follows the IMRAD logic for developing your introduction

The first sentences of intro is a general statement fact cant be supported by a ref in Ebola ! please change reliable text about.

Change it: Merrell LK, Blackstone SR. Women's Empowerment as a Mitigating Factor for 439 Improved Antenatal Care Quality despite Impact of 2014 Ebola Outbreak in Guinea. 440 Int J Environ Res Public Health. 2020;17(21):8172. 441 https://doi.org/10.3390/ijerph17218172. PMID: 33167397

Ref [2] (Liu et al., 2024) is a scoping review of instruments measuring empowerment in pregnant women. That is not the primary WHO policy text.

Ref [3] (Borghei et al., 2016) is a tool development/validation study in Iran. It is acceptable only if the sentence is about measuring empowerment or if the focus is explicitly Iran or instrument validation. For a general statement about what empowerment enables, cite conceptual reviews (e.g., concept analyses or global reviews)

Ref [9] is Bayati et al., 2018, an Iran study about health literacy education in a health center. That is not a primary source for the WHO/UHC conceptual claim. This is exactly the mismatch you noted: a local interventional/education study cannot be used as the authority for what UHC’s goals are. The general universal statement “Another goal of universal health coverage is to empower individuals in the area of self-care 88 [9].”

refers to a interlay study in Iran ! need to be re change

Ref [11] (Khayat et al., 2022) is a quasi-experimental telemedicine vs face-to-face training study. That paper may describe self-care operationally in the study, but if the sentence is a definition, a more authoritative definitional source (WHO self-care definition or a commonly used conceptual article) would be preferable. If Khayat et al. actually provides a clear definition, clarify that the citation is an example of an operational definition used in empirical studies.

Ref [12] (Rezaei et al., 2025) is a BMC Pregnancy Childbirth article (family medicine program) — appropriate for demonstrating importance in a vulnerable or covered population if the study actually examines access barriers. But again, for the broad claim “especially crucial,” consider adding a policy or review citation showing access barriers for pregnant women (WHO, Lancet Global Health, or systematic reviews).

Ref [7] is a midwife-oriented group counseling clinical trial in Shiraz — it is fine to cite as direct empirical evidence but do not over-generalize from a single trial.

Studies conducted in Iran among pregnant women indicate that approximately 30–40% … limited health literacy [20, 21].Refs [20] and [21] are listed as 2025 cross-sectional studies — those are appropriate for a regional/statistical claim. Ensure that the percentages quoted are exactly reported in those studies and that the sampling frame matches (pregnant women, same setting).

Method

Blinding

Well-explained why full blinding is impossible. Could cite a methodological ref (e.g., Schulz & Grimes, Lancet, 2002) for handling blinding in behavioral RCTs.

Discussion

You mention “integration of health literacy principles” and “virtual mode increases access,” but mechanisms are not deeply discussed (e.g., cognitive engagement, social learning, self-efficacy theory). Needs theoretical depth.

Implied in a few sentences (“integrating virtual counseling into routine care”), but not developed. Needs a closing paragraph connecting results to maternal health programs, virtual interventions, or health literacy policy.

Only one brief mention (“further research needed to confirm long-term effects”). Should specify what kind of follow-up studies are recommended (e.g., longitudinal, multi-site, cost-effectiveness).

**Do you want your identity to be public for this peer review?** For information about this choice, including consent withdrawal, please see our Privacy Policy

Reviewer #1: No

Reviewer #2: No

---

## [Author Response · Author response to Decision Letter 1]

7 Nov 2025

Dear Editor

I would like to thank the valuable comments of the respected reviewers; the following have been applied.

Comments Response

Reviewer: 1

• Given that this research work has previously been published under a protocol in the journal BMJ Open with doi:10.1136/bmjopen-2024-097991, different parts of the manuscript should be reviewed and revised for any dissimilarity. • Thank you for your valuable feedback. I have taken your suggestion into account and revised the manuscript to minimize dissimilarities with the previously published protocol (BMJ Open, doi:10.1136/bmjopen-2024-097991). Specifically, I reviewed and revised sections that could overlap, ensuring conceptual consistency while avoiding verbatim duplication, and I clearly delineated the methods and results to reflect the current study as an original full report.

• I also removed Table 2. Outline of counseling intervention program which was in the protocol article.

• In Table 4, explain the details of the scores for the different dimensions of empowerment, self-care, and health literacy. Table 4 has been changed to Table 3 in the new version. As per your valuable suggestion, the range of scores has been explained below the table.

Provide a brief explanation of the statistically significant relationship between the characteristics listed in Table 3 with Empowerment, Self-care, and Health literacy outcomes in the Methods part and a comparison with other studies in the Discussion part. • It's done.

Reviewer: 2

• Introduction • The entire text of the introduction was revised.

• Method

Blinding

Well-explained why full blinding is impossible. Could cite a methodological ref (e.g., Schulz & Grimes, Lancet, 2002) for handling blinding in behavioral RCTs. • It's done.

• Discussion: It's done.

---

## [Decision Letter · Decision Letter 1]

11 Dec 2025

Dear Dr. Bakouei,

Thank you for submitting your manuscript to PLOS ONE. After careful consideration, we feel that it has merit but does not fully meet PLOS ONE’s publication criteria as it currently stands. Therefore, we invite you to submit a revised version of the manuscript that addresses the points raised during the review process.

Please submit your revised manuscript by  Jan 25 2026 11:59PM . If you will need more time than this to complete your revisions, please reply to this message or contact the journal office at plosone@plos.org . A rebuttal letter that responds to each point raised by the academic editor and reviewer(s). You should upload this letter as a separate file labeled 'Response to Reviewers'.A marked-up copy of your manuscript that highlights changes made to the original version. You should upload this as a separate file labeled 'Revised Manuscript with Track Changes'.An unmarked version of your revised paper without tracked changes. You should upload this as a separate file labeled 'Manuscript'.

We look forward to receiving your revised manuscript.

Kind regards,

Fatemeh Zarei, PhD

Academic Editor

PLOS One

Journal Requirements:

Additional Editor Comments:

Dear Dr. Bakosi,

Thank you for your efforts. Please review the comments from Reviewer 1 carefully, particularly the sections highlighted in red.

Reviewers' comments:

Reviewer's Responses to Questions

**Comments to the Author**

Reviewer #1: (No Response)

Reviewer #2: All comments have been addressed

2. Is the manuscript technically sound, and do the data support the conclusions?

Reviewer #1: Yes

Reviewer #2: Yes

3. Has the statistical analysis been performed appropriately and rigorously?

Reviewer #1: I Don't Know

Reviewer #2: I Don't Know

4. Have the authors made all data underlying the findings in their manuscript fully available?

Reviewer #1: No

Reviewer #2: Yes

5. Is the manuscript presented in an intelligible fashion and written in standard English?

Reviewer #1: Yes

Reviewer #2: Yes

**Reviewer #1: Dear corresponding author**

There is a misunderstanding about the comments. The comments were further explained.

About previous comment: explain the details of the scores for the different dimensions of empowerment, self-care, and health literacy: it means explain the details of the scores of empowerment domains such as self-efficacy, self-esteem, future image, and pleasure of adding a new member to the family, perception of support from others; and Self-care domains such as nutrition performance, personal hygiene, smoking and drug use, exercise and physical activity, and routine pregnancy care and also; five dimensions of Health literacy in the table. You haven't talked about the details of the domains' scores anywhere in the manuscript and have explained the scores in general.

About previous comment: Provide a brief explanation of the statistically significant relationship between the characteristics listed in Table 3 with Empowerment, Self-care, and Health literacy outcomes in the Methods part and a comparison with other studies in the Discussion part. It means analyze correlation between Characteristics listed in table (age, Level of education, job, family income…) with Empowerment, Self-care, and Health literacy scores in a table and Compare this correlation to other studies in the discussion part.

I am waiting for your answer. Good luck.

Reviewer #2: No reviewer comments were left, indicating that the authors’ efforts to improve the manuscript were satisfactory.

**Do you want your identity to be public for this peer review?** For information about this choice, including consent withdrawal, please see our Privacy Policy

Reviewer #1: No

Reviewer #2: **Yes:** Fatemeh Zarei

---

## [Author Response · Author response to Decision Letter 2]

23 Dec 2025

Dear Editor

I would like to thank the valuable comments of the respected reviewers; the following have been applied.

Comments Response

Reviewer: 1

Provide a brief explanation of the statistically significant relationship between the characteristics listed in Table 3 with Empowerment, Self-care, and Health literacy outcomes in the Methods part and a comparison with other studies in the Discussion part. It means analyze correlation between Characteristics listed in table (age, Level of education, job, family income…) with Empowerment, Self-care, and Health literacy scores in a table and Compare this correlation to other studies in the discussion part. We thank the reviewer for this thoughtful comment. The primary objective of the present study was to evaluate the effect of a health literacy–based virtual group counseling intervention on empowerment, self-care, and health literacy outcomes within a randomized controlled trial framework. Accordingly, the sample size was calculated and powered exclusively to detect between-group differences in these outcomes, rather than to examine associations between baseline sociodemographic characteristics and outcome measures.

Moreover, as shown in Table 2 (changed number in new version), randomization resulted in well-balanced intervention and control groups, with no statistically significant differences in baseline sociodemographic or fertility characteristics. In randomized controlled trials, such baseline characteristics are not considered confounders, and conducting additional correlation or association analyses may be statistically underpowered and potentially misleading when the study is not designed for that purpose.

For these reasons, and in line with methodological recommendations for randomized trials, we did not perform additional correlation analyses between baseline characteristics and outcome measures. Instead, we focused on estimating the intervention effect using generalized estimating equations, which appropriately account for within-subject correlations over time and provide robust estimates of group-by-time interactions.

Importantly, the significant effects observed for empowerment, self-care, and health literacy can be attributed to the intervention itself rather than to baseline differences between participants.

Explain the details of the scores for the different dimensions of empowerment, self-care, and health literacy: it means explain the details of the scores of empowerment domains such as self-efficacy, self-esteem, future image, and pleasure of adding a new member to the family, perception of support from others; and Self-care domains such as nutrition performance, personal hygiene, smoking and drug use, exercise and physical activity, and routine pregnancy care and also; five dimensions of Health literacy in the table. You haven't talked about the details of the domains' scores anywhere in the manuscript and have explained the scores in general. It's done. Following this comment, figures 2, 3, and 4 were added to the findings and some points were also added to the discussion.

---

## [Editor Report · Decision Letter 2]

26 Dec 2025

Effect of virtual group counseling based on health literacy on the empowerment and self-care of pregnant women: a randomized controlled trial

PONE-D-25-39496R2

Dear Dr. Fatemeh Bakouei

We’re pleased to inform you that your manuscript has been judged scientifically suitable for publication and will be formally accepted for publication once it meets all outstanding technical requirements.

Kind regards,

Fatemeh Zarei, PhD

Academic Editor

PLOS One
---

## [Editor Report · Acceptance letter]

PONE-D-25-39496R2

PLOS One

Dear Dr. Bakouei,

I'm pleased to inform you that your manuscript has been deemed suitable for publication in PLOS One. Congratulations! Your manuscript is now being handed over to our production team.

Kind regards,

on behalf of

Dr. Fatemeh Zarei

Academic Editor

PLOS One